# Associations between Neurocardiovascular Signal Entropy and Physical Frailty

**DOI:** 10.3390/e23010004

**Published:** 2020-12-22

**Authors:** Silvin P. Knight, Louise Newman, John D. O’Connor, James Davis, Rose Anne Kenny, Roman Romero-Ortuno

**Affiliations:** 1The Irish Longitudinal Study on Ageing (TILDA), School of Medicine, Trinity College Dublin, D02 R590 Dublin, Ireland; louise.newman@tcd.ie (L.N.); john.oconnor@qub.ac.uk (J.D.O.); davisj5@tcd.ie (J.D.); rkenny@tcd.ie (R.A.K.); romeroor@tcd.ie (R.R.-O.); 2Discipline of Medical Gerontology, School of Medicine, Trinity College Dublin, D02 R590 Dublin, Ireland; 3School of Medicine, Dentistry and Biomedical Sciences, The Patrick G Johnston Centre for Cancer Research, Queen’s University, Belfast BT9 7BL, UK; 4Mercer’s Institute for Successful Ageing (MISA), St. James’s Hospital, D08 NHY1 Dublin, Ireland; 5Global Brain Health Institute, Trinity College Dublin, D02 DK07 Dublin, Ireland

**Keywords:** approximate entropy, sample entropy, physical frailty, cardiovascular, neurovascular, blood pressure, heart rate, frontal lobe oxygenation, near infrared spectroscopy, NIRS, TILDA

## Abstract

In this cross-sectional study, the relationship between noninvasively measured neurocardiovascular signal entropy and physical frailty was explored in a sample of community-dwelling older adults from The Irish Longitudinal Study on Ageing (TILDA). The hypothesis under investigation was that dysfunction in the neurovascular and cardiovascular systems, as quantified by short-length signal complexity during a lying-to-stand test (active stand), could provide a marker for frailty. Frailty status (i.e., “non-frail”, “pre-frail”, and “frail”) was based on Fried’s criteria (i.e., exhaustion, unexplained weight loss, weakness, slowness, and low physical activity). Approximate entropy (ApEn) and sample entropy (SampEn) were calculated during resting (lying down), active standing, and recovery phases. There was continuously measured blood pressure/heart rate data from 2645 individuals (53.0% female) and frontal lobe tissue oxygenation data from 2225 participants (52.3% female); both samples had a mean (SD) age of 64.3 (7.7) years. Results revealed statistically significant associations between neurocardiovascular signal entropy and frailty status. Entropy differences between non-frail and pre-frail/frail were greater during resting state compared with standing and recovery phases. Compared with ApEn, SampEn seemed to have better discriminating power between non-frail and pre-frail/frail individuals. The quantification of entropy in short length neurocardiovascular signals could provide a clinically useful marker of the multiple physiological dysregulations that underlie physical frailty.

## 1. Introduction

Frailty can be defined as a biologically driven decrease in reserve and resistance to stressors, resulting from collective declines across multiple physiological systems, which causes increased vulnerability to adverse outcomes such as mortality, institutionalization, falls, and hospitalization [1,2,3,4]. In this study, we used the frailty phenotype as proposed by Fried et al. [1] to define non-frail, pre-frail, and frail groups. This model has been extensively used in clinical practice and research [5]. The phenotype is based on five components, namely, unintentional weight loss, self-reported exhaustion, weakness, slow walking speed, and low physical activity. By this operationalization, pre-frailty is defined (independently of age and sex) as the presence of one or two criteria, and frailty as having three or more [1]. Despite its long conceptual and operational history, the intrinsic dynamic physiological mechanisms of frailty are not well understood [6].

Dysregulation of the neurovascular and cardiovascular systems under conditions of stress have been shown to be associated with risk of frailty [7,8,9]. A simple method to stress the neurocardiovascular system in clinical practice is by asking a person to remain lying supine for a few minutes, and then asking him/her to stand quickly. This is generally referred to as the orthostatic “active stand” test. The active stand challenges the body’s ability to compensate for the natural drop in blood pressure that occurs after standing due to gravity, and humans may perform this up to 50 times per day [10]. Continuously monitoring cardiovascular and neurovascular activity (such as blood pressure, heart rate, and frontal lobe brain oxygenation levels) during this challenge can provide clinically useful information with regards a person’s ability to compensate and recuperate from the stressor of standing. Currently, there is no consensus as to the most appropriate way to analyze and interpret continuously-measured active stand data; however, the Irish Longitudinal Study on Ageing (TILDA) has pioneered research in this area, using both standard statistical methods [11,12], as well as advanced data-driven approaches [7,13].

Disorder in physiological signals can be assessed by means of entropy [14,15,16,17]. Entropy is essentially a measure of irregularity/unpredictability, assigning lower entropy values to periodic, predictable data, and higher entropy values to irregular, unpredictable data. Multiple different implementations of entropy have been proposed for the analysis of time-varying physiological signals, including approximate entropy (ApEn), sample entropy (SampEn), multi-scale entropy, and cross entropy [17,18,19,20]. In the present work, we investigated two of the most widely used entropy measures for investigating physiological time series data, namely, ApEn and SampEn. Having been initially developed for physiological applications, both ApEn and SampEn have been demonstrated to provide reliable estimates of signal complexity in cardiovascular data [14,17,18,19,21,22,23,24]. ApEn was first proposed in 1991 by Pincus et al. [25]. Briefly, given a time-series of length *N*, ApEn approximates the negative average logarithm of the conditional probability that two trajectories of length *m* remain similar in the next timestep, within a tolerance specified as ±*r* * standard deviation (SD) of the time-series. ApEn provides a unit-less number from 0 to 2. Notably, ApEn counts each subset as matching itself, and therefore, the ApEn algorithm is inherently biased towards regularity. In 2000, Richman and Moorman [17] introduced SampEn. Similar to ApEn, SampEn is defined as the negative natural logarithm of the conditional probability that two trajectories of length *m* remain similar for *m* + 1; however, self-matches are not considered in the probability calculation in this instance. Additionally, it has been demonstrated that SampEn is largely independent of the data length and can potentially provide more consistent results than ApEn [17]. In the present study, we utilized both ApEn and SampEn for the analysis of neurocardiovascular signal complexity.

To date, a handful of studies have used both ApEn and SampEn specifically for the analysis of cardiovascular signal complexity in the context of frailty [21,22,23,26]. However, previously reported results have been contradictory, with some studies reporting higher complexity for frail individuals [23,26], while others reporting a reduction in complexity for frail individuals [21,22]. This may be due to differences in type of physiological signals analyzed, frailty operationalization methodologies employed, and/or small sample sizes. 

Because frailty is associated with adverse health outcomes in older people, it is important to detect it as soon as possible before it manifests as a visible disability. Indeed, medical research has shown that interventions can delay and even reverse frailty, especially when it presents in the early stages [27,28]. For that purpose, it is important to consider a frailty measure that does not include disability in its definition, and in that regard, Fried’s physical frailty phenotype is not only suitable, but also one of the most widely used in clinical practice [5]. We hypothesized that a simple automated measure of neurocardiovascular signal complexity (entropy) could provide a clinically useful marker of the multiple physiological dysregulations that underlie physical frailty. Given the critical importance of the performance of cardiovascular and neurovascular systems in helping us deal with stressors, our aim was to explore the relationship between entropy in these physiological signals and physical frailty. In other words, we undertook to classify frailty groups on the basis of entropy measures (ApEn, SampEn) and other demographic and health variables.

## 2. Materials and Methods

### 2.1. Study Population

This cross-sectional study utilized data from TILDA, an ongoing nationally representative prospective cohort study of community-dwelling adults (representing approximately 1 in 150 individuals in Ireland aged ≥50 years) established in 2009 (*N* = 8507). TILDA’s study design and sampling methods are detailed elsewhere [29,30,31]. Briefly, sampling was based on geographic clustering. Ongoing health, social, and economic data collection involves a computer-assisted personal interview (CAPI) and a self-completed questionnaire (SCQ) approximately every 2 years. Every second wave, participants take part in a comprehensive health assessment at a dedicated health center. The primary exposure variables for this study were measured at Wave 3 of TILDA, which took place between March 2014 and December 2015. Ethical approval was granted from the Health Sciences Research Ethics Committee at Trinity College Dublin (granted 9 June 2014 for Wave 3; approval reference “Main Wave 3 Tilda Study”) and all participants provided written informed consent. All research was performed in accordance with the Declaration of Helsinki.

### 2.2. Neurocardiovascular Measurements

Participants began the assessment with the affixing of a digital photoplethysmograph to the middle finger of the left hand (Finometer MIDI device, Finapres Medical Systems BV, Amsterdam, the Netherlands). This arm was then placed in a sling (to discourage its use during transition from supine to stand), resulting in the measurement site on the finger being roughly at the level of the heart throughout (further height changes adjusted for using the built-in height sensor on the Finometer device). Beat-to-beat blood pressure and heart rate were measured at 200 Hz using the Finometer device. Cerebral oxygenation was also measured simultaneously at 50 Hz using a near-infrared spectroscopy (NIRS) device (Portalite; Artinis Medical Systems, Zetten, the Netherlands) that was fixed to the forehead in approximately the FP1 (left frontal) position of the 10 to 20 electrode system (3 cm lateral and 3.5 cm superior to the nasion) [32]. This NIRS device uses 3 transmitters and 1 receiver, with each transmitter emitting 2 different wavelengths of light (760 nm and 850 nm) that propagate through the skull to a depth of approximately 2–3 cm and are absorbed at different rates by oxygenated haemoglobin (O2Hb) and deoxygenated haemoglobin (HHb). Hence, serum concentration levels of these molecules can be measured on the basis of the principle of absorption of electromagnetic radiation as described by the modified Beer–Lambert law. Multiple transmitters enable absolute concentration values to be determined via spatial resolved spectroscopy [33]. O2Hb and HHb concentrations were recorded, and tissue saturation index (TSI) was calculated as the O2Hb value expressed as a percentage of the sum of O2Hb and HHb values. The influence of environmental light was minimized via a black headband covering the sensor. All measurements were carried out in a comfortably lit room at an ambient temperature between 21 and 23 °C. Participants laid supine for ≈10 min before transitioning to a standing position and remained standing still for 3 min while data were continuously recorded.

### 2.3. Signal Processing

Signals for systolic blood pressure (sBP), diastolic blood pressure (dBP), mean arterial pressure (MAP), heart rate (HR), O2Hb, HHb, and TSI were extracted using MATLAB (R2019b, TheMathWorks, Inc., Natick, MA, USA). For the resting state, data from the last minute of supine rest were utilized in this study (prior to 60 s active stand “baseline” recording). For the active stand data, we analyzed a 1-min section, starting from when the participant began their stand (“challenge”), and another 1-min section taken from 120 to 180 s post-stand was analyzed as “recovery” from the active stand challenge. Time-to-stand was determined using the Finometer device’s built-in height sensor [34]. All data were down sampled to 5 Hz prior to analysis, providing *N* = 300 data points for analysis, and no filtering was applied. These values were chosen to provide a data collection timeframe that is easily transferable for clinical use (60 s), has an appropriate number of data points for analysis (time series with <200 data points are not recommended for either ApEn or SampEn due to inadequate vector matching [24,35]), while still capturing physiologically relevant signal components. Stationarity of the data was assessed via augmented Dicky–Fuller tests on both the original raw data and transformed data (data were transformed by subtracting the mean and dividing it by the standard deviation to increase the stationarity of the data series [36]).

### 2.4. Entropy Analysis

Entropy analysis was performed using MATLAB. Previously developed MATLAB scripts were used to calculate ApEn and SampEn [15,16]. A detailed description of the algorithms used to compute both ApEn and SampEn has been previously reported in detail [17,25]; however, below we provide a brief overview.

#### 2.4.1. Approximate Entropy (ApEn)

ApEn [25] was calculated as
(1)$$\mathrm{ApEn}\left(m,r,N\right)≔\frac{1}{\left(N-m+1\right)}\overset{N-m+1}{\underset{i=1}{\sum }}\log\frac{C_{i}^{m}\left(r\right)}{C_{i}^{m+1}\left(r\right)},\;$$
where $C_{i}^{m}\left(r\right)$ is the number of points found within the distance r for any point *x*(*i*) within the points $x_{i}^{m}$: = [*x*(*i*),..., *x*(*i* + *m* − 1)], divided by *N* − *m* + 1. In this study, *m* (embedding dimension; the length of the data segment being compared) was set to 2 for both ApEn and SampEn, as this has been shown to show good statistical validity for ApEn and SampEn, especially for biological data [14,35]. The effects of increasing *m* were also investigated, with results from *m* = 2, 3, and 4 presented comparatively in Appendix B. An optimal *r* (similarity criterion) was computed to give the maximum ApEn using *r* from 0 to 0.6 in increments of 0.02, as per the method proposed by Chon et al. [14].

#### 2.4.2. Sample Entropy (SampEn)

SampEn [17] was calculated as
(2)$$\mathrm{SampEn}\left(m,r,N\right)≔\log\left(\overset{N-m}{\underset{i=1}{\sum }}C_{i}^{m}\left(r\right)\right)-\log\left(\overset{N-m-1}{\underset{i=1}{\sum }}C_{i}^{m+1}\left(r\right)\right),\;$$
however, in this instance, $C_{i}^{m}\left(r\right)$ does not count self-matches. For SampEn, an *r* of 0.15 was selected, in line with previous recommendations for similar physiological data [37,38]. To assess the effects of data stationarity on entropy measures, we calculated ApEn and SampEn for both the original raw and transformed data.

### 2.5. Frailty Phenotype

The calculation of the frailty phenotype was conducted following the methodology proposed by Fried et al. [1]. Full details have been described previously [39,40,41]; briefly, the frailty phenotype was operationalized using population-specific cut-off points related to differences in the assessments of weakness (sex- and body mass index-adjusted grip strength measured with dynamometer on the dominant hand), physical activity (sex-adjusted kilocalories from the International Physical Activity Questionnaire—Short Form [42]), and walking speed (sex- and height-adjusted cm per second using the GAITRite portable walkway (CIR Systems, Inc., Sparta, NJ, USA)). Weight loss was determined by the question “In the past year have you lost 10 pounds (4.5 kg) or more in weight when you were not trying to?” Exhaustion was captured using 2 items from the Centre for Epidemiological Studies Depression (CESD) scale [43]. Participants were asked how often they felt that “I could not get going” and “I felt that everything I did was an effort”. A response of “moderate amount/all of the time” to either question was considered as “exhaustion”.

### 2.6. Other Measures

As part of the TILDA assessment, the following self-reported measures were also recorded at Wave 3 of the study: educational attainment, cardiovascular conditions (angina, high blood pressure, heart failure, heart murmur, abnormal heart rhythm, heart attack (ever), stroke (ever), or transient ischemic attack (TIA, ever)), diabetes, alcohol consumption habits (CAGE) [44], smoking history, and anti-hypertensive medication use (coded using the Anatomical Therapeutic Chemical Classification (ATC): antihypertensive medications (ATC C02), diuretics (ATC C03), β-blockers (ATC C07), calcium channel blockers (ATC C08), and renin-angiotensin system agents (ATC C09)). Depressive symptoms were assessed using the CESD scale [45]. Time taken to stand was calculated using data from the Finometer device’s built-in height sensor, as previously described [34]. To describe the general level of disability, we recorded the number of difficulties in performing activities of daily living (ADL). The original ADL scale, developed by Katz et al., encompasses “activities which people perform habitually and universally”, such as dressing (including putting on shoes and socks), walking across a room, bathing or showering, eating (such as cutting up food), getting in or out of bed, and using the toilet (including getting up and down) [46].

### 2.7. Statistical Analysis

Statistical analysis was performed using STATA 14.1 (StataCorp, College Station, TX, USA). The data were visually assessed for normality via Q-Q plots and histograms. All multivariate analysis was completed using robust linear regression with residual analysis completed to assess model assumptions. Statistical significance was set at *P* < 0.05. Multiple models were utilized to examine the relationships between the neurocardiovascular entropy measures and frailty phenotype status. Additional potential correlates controlled for in all models were age, sex, education, number of cardiovascular conditions (0, 1, 2+), diabetes, antihypertensive medication, alcohol consumption habits, smoking, and depression. The models for active stand data additionally controlled for stand time. Results from absolute coefficients were reported as point estimates in appropriate units, presented with 95% confidence intervals (CI). 

### 2.8. Data Availability Statement

The datasets generated during and/or analyzed during the current study are not publicly available due to data protection regulations but are accessible at TILDA on reasonable request. The procedures to gain access to TILDA data are specified at https://tilda.tcd.ie/data/accessing-data/.

## 3. Results

### 3.1. Participant Characteristics

In total 2645 participants had complete blood pressure (BP) resting state and active stand data and 2225 had full NIRS data; full exclusions leading to these cohorts are provided in Figure 1. In total 2172 individuals had both BP and NIRS data. Participants’ mean (SD) age was 64.3 (7.7) years in both samples. In the BP cohort, 53.0% were female and in the NIRS cohort 52.3% were female. Similar distributions of frailty phenotype status were seen across both cohorts (non-frail: 59.1% (BP), 59.5% (NIRS); pre-frail: 37.2% (BP), 36.7% (NIRS); frail 3.7% (BP), 3.8% (NIRS)). Full demographic characteristics for both cohorts are presented in Table 1.

### 3.2. Associations of Entropy with Frailty Phenotype

Figure 2 visually illustrates data from three example participants with “low” (0.20), “medium” (0.45), and “high” (0.70) levels of ApEn in their resting state sBP data. As was generally the case, individuals with higher entropy in one measure (in the case of Figure 2 sBP) generally had higher entropy in the other physiological measures investigated. Figure 2 illustrates this well, with the stratification of signal disorder still visually apparent in the frontal lobe NIRS measures (O2Hb, HHb, and TSI).

Results from augmented Dicky–Fuller tests revealed low proportions of stationarity (0.2% to 23.1%) for the raw data; after transforming the data, 18.1% to 92.6% of cases were stationary. Despite the increase in stationarity, which in some instances was large (e.g., resting state O2Hb stationarity increased from 0.2% to 92.6%), no differences in ApEn and SampEn measures were found between the original and transformed data (see Appendix A, Table A1). Thus, the results from the original time series were used for statistical analysis.

Table 2 and Table 3 provide the mean, SD, and range for ApEn and SampEn calculated from resting state, active stand (stand 0–60s), and recovery (120–180s post-stand), stratified by physical frailty status. Across all measures, absolute mean entropy was higher in pre-frail and frail groups compared with the non-frail group. Mean sBP, dBP, MAP, and HR entropy measures were highest during recovery and lowest during stand. For O2Hb, HHb, and TSI mean values increased from resting state to stand to recovery. Overall, absolute BP and HR SampEn measures were 3% to 50% lower than ApEn measures. Similarly, absolute SampEn was 6% to 36% lower than ApEn measures for NIRS resting state and active stand measures; however, absolute SampEn values in NIRS recovery were higher than ApEn measures. Of note, SD values reported in Table 2 and Table 3 are absolute and are not statistically adjusted for any confidence level.

Figure 3 reports multivariate-adjusted point estimates from the regression models, with error bars showing the 95% CIs corresponding to the 95% confidence level (i.e., *P* ≤ 0.05 considered significant). Overall, models controlled only for age and sex provided similar results to the fully controlled models (controlling for age, sex, education, number of cardiovascular conditions (0, 1, 2+), diabetes, antihypertensive medication, alcohol consumption habits, smoking, and depression). Models for active stand data also controlled for stand time). Beta coefficients were slightly lower for some of the measures in the fully controlled versus age and sex-controlled models, however, significance of the results was generally consistent, with the exception of the O2Hb and HHb resting state data, which were no longer significant in the fully controlled models. The magnitude of the beta coefficients from the regression analysis were in line with the absolute differences reported in Table 2 and Table 3.

In fully controlled models of resting state data, all four Finometer measures were significantly higher in pre-frail (ApEn—sBP: *β* = 0.01, *P* = 0.004; dBP: *β* = 0.01, *P* ≤ 0.001; MAP: *β* = 0.01, *P* ≤ 0.001; HR: *β* = 0.01, *P* = 0.049. SampEn—sBP: *β* = 0.01, *P* = 0.050; dBP: *β* = 0.02, *P* = 0.001; MAP: *β* = 0.02, *P* = 0.001; HR: *β* = 0.01, *P* = 0.038) and frail groups (ApEn—sBP: *β* = 0.02, *P* = 0.006; dBP: *β* = 0.04, *P* = 0.001; MAP: *β* = 0.04, *P* ≤ 0.001; HR: *β* = 0.03, *P* = 0.007. SampEn—sBP: *β* = 0.05, *P* = 0.004; dBP: *β* = 0.05, *P* = 0.003; MAP: *β* = 0.06, *P* ≤ 0.001; HR: *β* = 0.04, *P* = 0.048). 

For the fully controlled stand 0–60s models, all BP/HR entropy measures were significantly associated with frailty status (pre-frail: ApEn—sBP: *β* = 0.01, *P* = 0.001; dBP: *β* = 0.01, *P* = 0.001; MAP: *β* = 0.01, *P* = 0.001; HR: *β* = 0.01, *P* ≤ 0.001. SampEn—sBP: *β* = 0.02, *P* = 0.004; dBP: *β* = 0.01, *P* = 0.020; MAP: *β* = 0.01, *P* = 0.018; HR: *β* = 0.02, *P* = 0.012. frail: ApEn—sBP: *β* = 0.03, *P* = 0.003; dBP: *β* = 0.03, *P* = 0.001; MAP: *β* = 0.03, *P* = 0.003; HR: *β* = 0.04, *P* ≤ 0.001. SampEn—sBP: *β* = 0.04, *P* = 0.011; dBP: *β* = 0.06, *P* = 0.001; MAP: *β* = 0.04, *P* = 0.010; HR: *β* = 0.09, *P* ≤ 0.001).

Likewise, for the fully controlled stand 120–180s models, again all BP/HR entropy measures were significantly associated with frailty status (pre-frail: ApEn—sBP: *β* = 0.01, *P* = 0.004; dBP: *β* = 0.01, *P* = 0.008; MAP: *β* = 0.01, *P* = 0.006; HR: *β* = 0.01, *P* ≤ 0.001. SampEn—sBP: *β* = 0.01, *P* = 0.010; dBP: *β* = 0.01, *P* = 0.028; MAP: *β* = 0.02, *P* = 0.007; HR: *β* = 0.02, *P* = 0.001. Frail: ApEn—sBP: *β* = 0.03, *P* = 0.001; dBP: *β* = 0.04, *P* ≤ 0.001; MAP: *β* = 0.03, *P* = 0.001; HR: *β* = 0.04, *P* = 0.003. SampEn—sBP: *β* = 0.05, *P* = 0.003; dBP: *β* = 0.07, *P* ≤ 0.001; MAP: *β* = 0.06, *P* = 0.001; HR: *β* = 0.06, *P* = 0.004).

For the fully controlled models, NIRS (O2Hb, HHb, and TSI) entropy measurements were not significantly associated with frailty status, with the exception of TSI at resting state, which was higher in the pre-frail group (ApEn: *β* = 0.01, *P* = 0.011; SampEn: *β* = 0.01, *P* = 0.007); and in the recovery data, TSI entropy was significantly higher for frail participants (ApEn: *β* = 0.03, *P* = 0.010; SampEn: *β* = 0.10, *P* = 0.022). For all models investigated, the magnitude of statistically significant differences between frailty groups were larger for SampEn compared with ApEn, with *β* coefficients up to 102% higher in BP and 198% higher in NIRS SampEn results compared with ApEn.

The effect of increasing *m* was also investigated, with multivariate-adjusted point estimates from the regression models for *m* = 2, 3, and 4 presented in Appendix B, Figure A1. For BP and HR, all significant associations described above remained significant as *m* was increased. However, the associations of frail status with recovery TSI lost significance as *m* was increased.

## 4. Discussion

Results from this study demonstrate significant associations between peripherally measured neurocardiovascular signal entropy and physical frailty status. Even though the magnitude of these associations was shown to be similar for resting state, active stand, and recovery data, the differences between non-frail and pre-frail/frail BP and HR entropy measures did increase during the stand and recovery phases, most notably for HR. For frail individuals, TSI was significantly higher during the recovery from stand, compared with non-frail. Even though, overall, absolute SampEn values were 2 to 50% lower than ApEn values, while *β* coefficients from statistically significant models were up to 198% higher when using SampEn, which suggests potentially better discriminating power between non-frail and pre-frail/frail individuals for SampEn. These results support the hypothesis that a simple automated measure of neurocardiovascular signal entropy could provide a clinically useful marker of the multiple physiological dysregulations that underlie physical frailty. 

Only a handful of smaller scale studies have investigated the associations of cardiovascular signal entropy with frailty. Results reported to date are contradictory, with some reporting higher levels of entropy and disorder in cardiovascular data for pre-frail and frail individuals versus non-frail [23,26], while others contrariwise report lower entropy values in these groups [21,22]. However, there are several important methodological differences between these studies, as well as between those and the present work, which most likely account for this. Most similar to the present work in terms of methodology, and reporting similar results, Takahashi et al. [23], in a study on 80 individuals, found higher ApEn and conditional entropy for both pre-frail and frail groups, compared with non-frail, with frail participants having the highest entropy overall; in addition, the authors also reported lower absolute entropy in their stand data versus rest, which is also in line with our results, as well as other previous work [47]. Analogously, some studies have reported a reduction in entropy after a head-up tilt test [48,49], which is another type of orthostatic challenge, though less representative of daily living. Exploring the physiological origins of these signal complexity differences between resting state and orthostatic challenge would be of interest, particularly in relation to baroreflex control, which has been shown to be associated with frailty status [50]. Conversely, Chaves et al. indicated that lower values of ApEn were associated with a higher probability of an individual being frail [21]. Chaves et al. operationalized the original Fried’s criteria; however, direct comparison between that study and the present is not possible due to large difference in the methods used; for example, only females were investigated (*N* = 389), and a dichotomous classification of frailty was adopted (frail or non-frail). Most notably, their study utilized a much longer dataset (2–3 h), recorded while participants underwent transitions through a number of diverse postural positions (e.g., lying, sitting, standing). As such, their study is likely to be reporting on the flexibility of the cardiovascular system, i.e., the ability of the system to adapt to multiple challenges over a longer time period, a measure that is known to be indicative of a more positive health status. Another recent study by Rangasamy et al. likewise reported lower entropy in frail versus non-frail (*N* = 364) [22]. However, again methodological differences do not allow for direct comparison with the present study, including in this case the vastly different process used to quantify the dichotomous frailty status used (which was based on demographics, anthropometrics, and blood biomarkers). 

Heart rate variability complexity is generally expected to decrease with pathology [51]; however, in the present work, higher entropy in both BP and HR were found to be associated with increased frailty. We postulate that entropy calculated in short length neurocardiovascular data, as reported herein, is not measuring system flexibility, but rather systemic disorder, or “jitter”, resulting from dysregulation of the neurocardiovascular system. It would therefore be expected that this negative state of higher disorder on a short time frame would be associated with physical frailty or systemic dysregulation. One possible cause for this dysregulation could be an increase of sympathetic activity and/or modulation directed to the heart and/or blood vessels with increased frailty status, as previously described in abnormal ageing states [52]. Other potential influencing factors could be modified cardiac reserve, changes in arterial structure (e.g., increased stiffness, decreased compliance, and endothelial dysfunction), as well as changes of diastolic filling and increased collagen in the left ventricle [47]. Increasing the embedded dimension (*m* = 2, 3, and 4) did not have a major effect on the main significant results of this study (see Appendix B, Figure A1). This suggests that the patterns of increased disorder associated with frailty status in BP and HR data occur within the scale of 0.4 to 0.8 s.

The present study has several strengths. To date, ours is the largest study to investigate the associations between entropy measures in neurocardiovascular signals and physical frailty status (*N* = 2225/2645). Additionally, to the best of our knowledge, this is the first study to examine frontal lobe oxygenation entropy (as measured using NIRS) with the physical frailty phenotype. The rich data available as part of TILDA meant that models could be comprehensively controlled for a number of covariates known to affect physical and neurocardiovascular function. Additionally, the richness of the continuously, simultaneously measured neurovascular and cardiovascular data allowed for the assessment of several physiological measurements, recorded within the same experimental paradigm. 

From a clinical relevance point of view, it is important to notice the generally low levels of disability in this sample, with almost 80% of frail participants not having any ADL impairments (Table 1). Indeed, Fried’s physical frailty phenotype intends to capture a pre-disability state [53]. That, coupled with the fact that less than 4% of the sample were classified as frail, highlights the remarkable sensitivity of entropy measures in automatically identifying frailty status at resting state, when often it is very difficult for clinicians to identify frailty with the naked eye.

The methodologies presented herein were specifically designed to be highly transferable for use in a clinical setting. All measures were non-invasive and non-ionizing. The short data length required (60 s) would be feasible and practical for use in a busy clinic. Entropy provides a single-number measure, which could theoretically be calculated at and displayed on the measurement device itself, allowing for easy use by clinicians. Additionally, since the associations of entropy levels in resting state, active stand, and recovery data with frailty phenotype were all similar, this suggests that resting state entropy might be sufficient as a clinical marker for frailty, further increasing the ease by which this measure could be recorded in the clinic. Moreover, the similarity between BP and HR entropy measures, in relation to frailty status, suggests that these measures may provide complementary information, as has been previously reported [54], and as such a univariant approach (i.e., the assessment of one of these measures) may be sufficient for clinical use. Input parameters and implementation of ApEn and SampEn calculations were based on recommendations for similar physiological data from previous studies (*m* = 2 (also reported in Appendix B
*m* = 3, 4) [14,35]; *r* = 0.15 (SampEn) [37,38], optimum calculated [0 to 0.6] (ApEn) [14]; *N* > 200 [24,35]); however, a consensus with regards the optimal methodologies to use, as well as normative age- and sex-adjusted reference values, would be required for widespread clinical adoption. Further work is necessary to establish the prognostic implications of entropy measures vis-à-vis other clinical markers (e.g., for the prediction of mortality and other adverse health events). Future longitudinal work investigating how these measures vary over time would also be of interest, since this may provide an “early warning” measure for potential transitions from less to more adverse frailty statuses, for use in a clinical setting.

There are several further limitations to this study that should be kept in mind when interpreting the results. Analyses were cross-sectional and, as such, causality or even temporality of the observed relationships could not be inferred. There was a small number of frail individuals compared with the other groups; however, the proportion of frail individuals was in line with previously reported TILDA studies [55,56]. This slight discrepancy in the proportion of frail persons may have been due to the rigor of the data quality exclusion criteria used to ensure high internal validity of the study. Since a convenience sample was used in the current work (i.e., participants without full neurovascular or cardiovascular data and physical frailty data could not be included), we do not propose these results are population-representative, despite the large cohort sizes; it is reasonable to assume that a higher proportion of participants unable to provide these data (particularly active stand data) may have been from the frail group. 

Data utilized in this study had relatively high proportions of non-stationarity, as is commonly the case with physiological data [57], which may have potentially biased the estimates of complexity since non-stationarities have been shown to diminish the absolute level of complexity as assessed by conditional entropy [58]. However, there was no difference between entropy measures from the original raw data and data transformed to increase stationarity, even though stationarity increased by up to 92% in some instances, which is in line with previous work [35]. Further work exploring different methods to increase stationarity of these types of data, while still retaining the physiologically relevant signal complexity information, would be of interest. Another important caveat to keep in mind when interpreting the results of this study is that the ApEn results presented may be biased towards regularity, since ApEn counts self matches. Methods have been proposed to correct this bias in ApEn measures [48]; however, this was not done in the present study as we also present SampEn results in parallel, which are not subject to the same potential self-matching bias. Additionally, due to the fact that SampEn displays relative consistency under conditions where ApEn does not (e.g., data length) [17], it would be recommended to use the SampEn results from this study for future cross-study comparisons. In the present study, ApEn and SampEn were used to investigate each neurocardiovascular measure individually and at a single scale; future work using other entropy methods, such as multi-scale and cross entropy, would be of interest. Finally, this study does support the need for future longitudinal work to determine the clinical significance of these findings, and as such, this study should be considered preliminary and exploratory.

## 5. Conclusions

Results from this study demonstrated significant associations between peripherally measured neurovascular/cardiovascular signal entropy and physical frailty status. These results support the hypothesis that a simple automated measure of neurocardiovascular signal entropy at rest could provide a clinically useful marker of physical frailty. Our future work will focus on the study of physiological signal entropy as an early marker of the physiological dysregulation seen in frailty, which may open the possibility to detect early physiological dysregulation before the onset of obvious physical disability, and the opportunity to translate this work into opportunities to improve physiological resilience in older adults. 

## Figures and Tables

**Figure 1 entropy-23-00004-f001:**
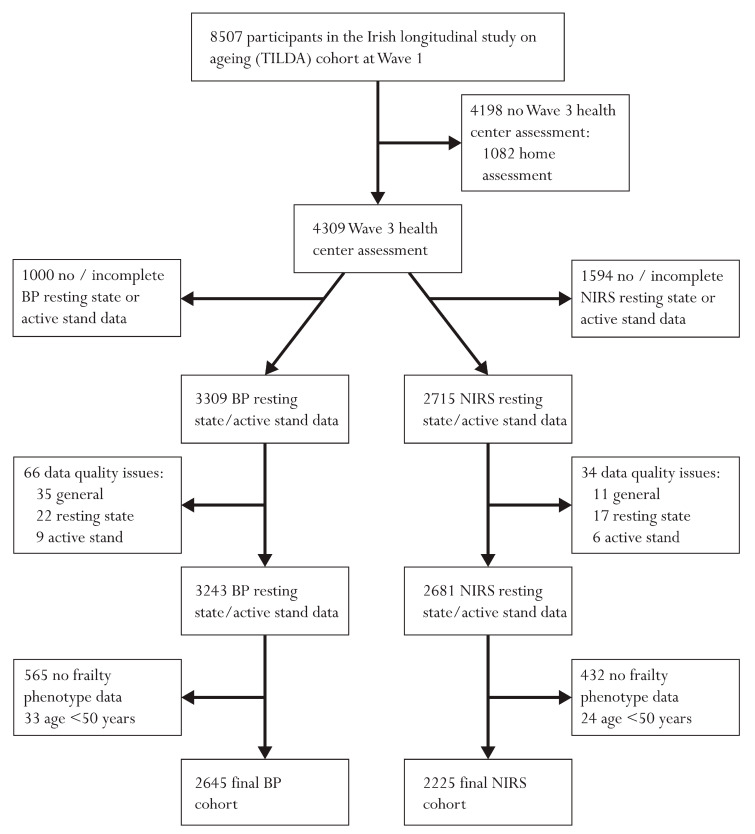
Flow chart describing sample selection and exclusions.

**Figure 2 entropy-23-00004-f002:**
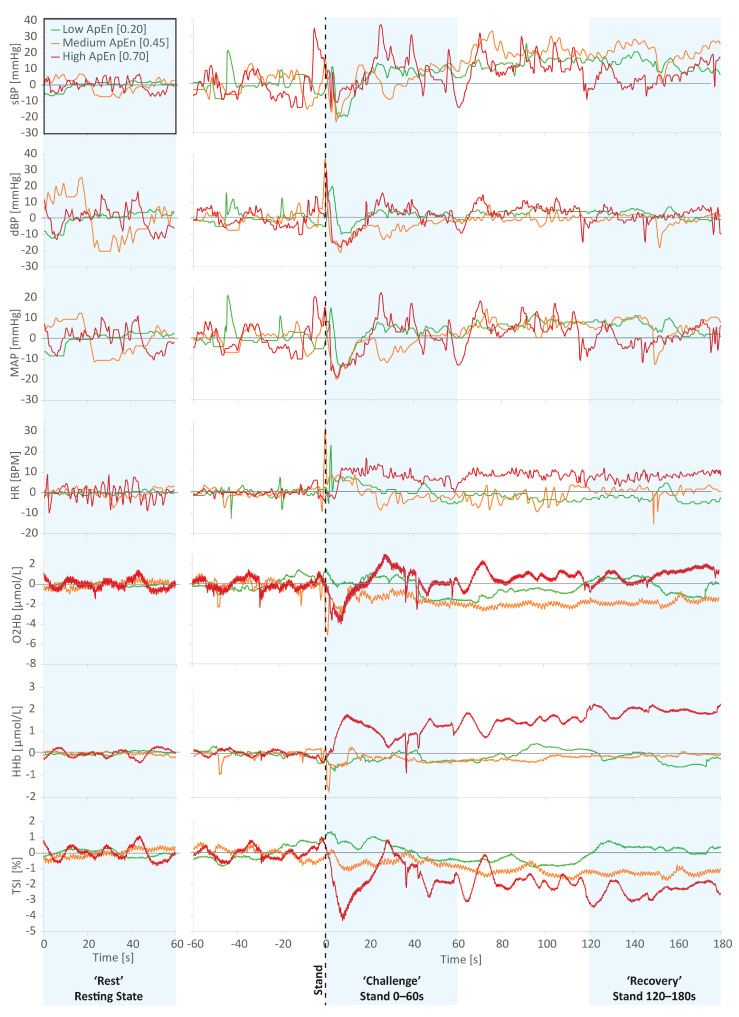
Example plots of data from three different participants with “low” (0.20), “medium” (0.45), and “high” (0.70) levels of approximate entropy (ApEn) measured in the resting state systolic blood pressure (sBP) data. Individuals with higher entropy in one measure investigated (in this case, resting state sBP) also generally had higher entropy in the other physiological measures investigated, as visually illustrated above.

**Figure 3 entropy-23-00004-f003:**
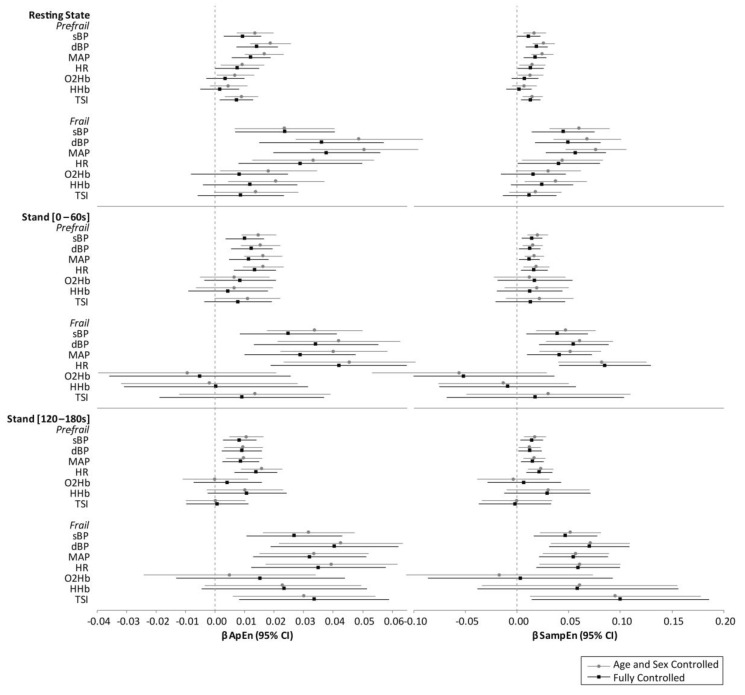
Results from robust multivariate linear regression showing beta coefficients (β) for both approximate entropy (ApEn) and sample entropy (SampEn). All models adjusted for age, sex, education, diabetes, number of cardiovascular conditions, antihypertensive medication use, alcohol consumption habits, smoking, and depression. Model for active stand data additionally controlled for stand time. Abbreviations: sBP: systolic blood pressure; dBP: diastolic blood pressure; MAP: mean arterial pressure; HR: heart rate; O2Hb: oxygenated hemoglobin concentration; HHb: deoxygenated hemoglobin concentration; TSI: tissue saturation index; β: beta coefficient.

**Table 1 entropy-23-00004-t001:** Demographic characteristics of the study samples.

	BP Cohort (*n* = 2645)	NIRS Cohort (*n* = 2225)
Age [years]	64.3 (SD: 7.7, range: [50–93])	64.3 (SD: 7.7, range: [50–93])
Sex [% (n)]	Female: 53.0% (1401)	Female: 52.3% (1163)
Education [% (n)]		
Primary/none	16.5% (436)	16.5% (368)
Secondary	39.9% (1055)	40.1% (891)
Third/higher	43.6% (1154)	43.4% (966)
Frailty Phenotype [% (n)]		
Non-frail	59.1% (1564)	59.5% (1325)
Pre-frail	37.2% (984)	36.7% (816)
Frail	3.7% (97)	3.8% (84)
Disability [% (n)]		
Number of ADL disabilities		
Non-frail		
0	98.9% (1547)	98.9% (1310)
1	1.0% (15)	1.0% (13)
+2	0.1% (2)	0.1% (2)
Pre-frail		
0	96.1% (946)	96.3% (786)
1	2.3% (23)	2.1% (17)
+2	1.6% (15)	1.6% (13)
Frail		
0	79.4% (77)	78.6% (66)
1	13.4% (13)	15.5% (13)
+2	7.2% (7)	5.9% (5)
No. of Cardiovascular Conditions ^a^ [% (n)]		
0	40.4% (1069)	40.6% (904)
1	35.6% (942)	35.9% (768)
2+	24.0% (634)	23.5% (523)
Self-reported diabetic [%]	6.8% (179)	6.5% (144)
Antihypertensive medications ^b^ [% (n)]	37.7% (997)	37.2% (827)
CAGE alcohol scale		
CAGE < 2	76.7% (2029)	77.2% (1718)
CAGE ≥ 2	12.4% (328)	12.3% (274)
No response	10.9% (288)	10.5% (233)
Smoker [% (n)]		
Never	47.9% (1268)	48.0% (1067)
Past	42.6% (1127)	42.5% (945)
Current	9.5% (250)	9.5% (213)
CESD [% (n)]		
Non-depressed (CESD < 9)	89.1% (2358)	89.3% (1986)
Depressed (CESD ≥ 9)	10.9% (287)	10.7% (239)
Time to stand [seconds]	7.2 (SD: 2.8, range: [2–27])	7.2 (SD: 2.8, range: [2–26])

^a^ Cardiovascular conditions: angina, high blood pressure, heart failure, heart murmur, abnormal heart rhythm, heart attack (ever), stroke (ever), or transient ischemic attack (TIA, ever). ^b^ Coded using the Anatomical Therapeutic Chemical Classification (ATC): antihypertensive medications (ATC C02), diuretics (ATC C03), β-blockers (ATC C07), calcium channel blockers (ATC C08), and renin–angiotensin system agents (ATC C09). Abbreviations: ADL, activities of daily living; CESD, Center for Epidemiologic Studies Depression scale.

**Table 2 entropy-23-00004-t002:** Approximate entropy (ApEn) results by frailty phenotype grouping.

ApEn	Non-Frail (*N* = 1325–1564)	Pre-Frail (*N* = 816–984)	Frail (*N* = 84–97)
Resting State	Mean (SD, [Range])	Mean (SD, [Range])	Mean (SD, [Range])
sBP	0.52 (0.07, [0.26–0.98])	0.54 (0.08, [0.22–0.77])	0.56 (0.08, [0.31–0.75])
dBP	0.45 (0.08, [0.20–0.96])	0.47 (0.09, [0.22–0.78])	0.50 (0.11, [0.29–0.83])
MAP	0.46 (0.08, [0.17–1.01])	0.48 (0.08, [0.22–0.79])	0.52 (0.09, [0.30–0.80])
HR	0.49 (0.09, [0.02–0.81])	0.49 (0.09, [0.05–0.77])	0.51 (0.10, [0.29–0.77])
O2Hb	0.44 (0.07, [0.16–0.68])	0.45 (0.07, [0.18–0.65])	0.46 (0.07, [0.23–0.60])
HHb	0.43 (0.07, [0.09–0.65])	0.43 (0.07, [0.16–0.64])	0.45 (0.07, [0.27–0.60])
TSI	0.39 (0.06, [0.12–0.62])	0.40 (0.07, [0.11–0.63])	0.41 (0.06, [0.28–0.54])
Stand (0–60s)			
sBP	0.44 (0.07, [0.20–0.97])	0.46 (0.08, [0.21–0.80])	0.48 (0.08, [0.33–0.70])
dBP	0.40 (0.07, [0.17–0.90])	0.42 (0.09, [0.19–0.83])	0.45 (0.10, [0.25–0.76])
MAP	0.41 (0.08, [0.17–0.93])	0.43 (0.08, [0.18–0.82])	0.45 (0.09, [0.30–0.80])
HR	0.45 (0.08, [0.07–0.80])	0.46 (0.09, [0.18–0.95])	0.49 (0.11, [0.23–0.81])
O2Hb	0.79 (0.12, [0.36–1.37])	0.80 (0.12, [0.32–1.34])	0.79 (0.14 [0.38–1.08])
HHb	0.69 (0.14, [0.21–1.37])	0.70 (0.15, [0.30–1.36])	0.69 (0.14, [0.40–0.99])
TSI	0.77 (0.12, [0.40–1.39])	0.79 (0.12, [0.41–1.33])	0.79 (0.12, [0.53–1.08])
Stand (120–180s)			
sBP	0.55 (0.07, [0.29–1.04])	0.56 (0.07, [0.33–0.84])	0.58 (0.07, [0.39–0.78])
dBP	0.49 (0.08, [0.27–1.02])	0.50 (0.09, [0.26–0.85])	0.53 (0.10, [0.32–0.84])
MAP	0.51 (0.07, [0.22–1.01])	0.52 (0.08, [0.26–0.81])	0.54 (0.09, [0.28–0.84])
HR	0.49 (0.08, [0.02–0.88])	0.50 (0.09, [0.23–0.85])	0.52 (0.11, [0.20–0.79])
O2Hb	0.91 (0.12, [0.43–1.40])	0.91 (0.12, [0.55–1.37])	0.92 (0.13 [0.50–1.12])
HHb	0.87 (0.14, [0.40–1.42])	0.88 (0.14, [0.42–1.36])	0.90 (0.11, [0.62–1.18])
TSI	0.92 (0.11, [0.50–1.34])	0.92 (0.11, [0.59–1.34])	0.95 (0.11, [0.60–1.15])

Abbreviations: sBP: systolic blood pressure; dBP: diastolic blood pressure; MAP: mean arterial pressure; HR: heart rate; O2Hb oxygenated hemoglobin concentration; HHb deoxygenated hemoglobin concentration; TSI: tissue saturation index.

**Table 3 entropy-23-00004-t003:** Sample entropy (SampEn) results by frailty phenotype grouping.

SampEn	Non-Frail (*N* = 1325–1564)	Pre-Frail (*N* = 816–984)	Frail (*N* = 84–97)
Resting State	Mean (SD, [Range])	Mean (SD, [Range])	Mean (SD, [Range])
sBP	0.49 (0.13, [0.07–1.41])	0.51 (0.14, [0.08–0.93])	0.55 (0.14, [0.15–0.87])
dBP	0.39 (0.13, [0.06–1.34])	0.41 (0.13, [0.06–0.95])	0.45 (0.16, [0.18–0.99])
MAP	0.40 (0.13, [0.05–1.27])	0.43 (0.14, [0.08–0.84])	0.48 (0.15, [0.15–0.89])
HR	0.44 (0.15, [0.01–1.00])	0.44 (0.16, [0.02–1.03])	0.46 (0.19, [0.08–0.99])
O2Hb	0.34 (0.14, [0.01–0.71])	0.35 (0.14, [0.05–0.78])	0.38 (0.14, [0.07–0.67])
HHb	0.31 (0.13, [0.03–0.80])	0.32 (0.14, [0.04–0.72])	0.36 (0.14, [0.12–0.62])
TSI	0.26 (0.10, [0.03–0.72])	0.27 (0.11, [0.03–0.74])	0.28 (0.11, [0.06–0.52])
Stand (0–60 s)			
sBP	0.30 (0.12, [0.04–1.26])	0.32 (0.13, [0.04–0.98])	0.35 (0.14, [0.12–0.82])
dBP	0.20 (0.11, [0.03–1.03])	0.22 (0.14, [0.04–1.03])	0.28 (0.16, [0.03–0.87])
MAP	0.24 (0.11, [0.04–1.11])	0.26 (0.13, [0.04–0.92])	0.30 (0.15, [0.05–0.99])
HR	0.31 (0.14, [0.02–1.08])	0.33 (0.17, [0.03–1.21])	0.40 (0.21, [0.03–1.05])
O2Hb	0.73 (0.37, [0.04–2.69])	0.74 (0.39, [0.01–2.70])	0.68 (0.38, [0.03–1.54])
HHb	0.44 (0.33, [0.02–2.88])	0.47 (0.35, [0.01–2.80])	0.44 (0.29, [0.03–1.30])
TSI	0.61 (0.36, [0.07–2.73])	0.64 (0.36, [0.06–2.63])	0.67 (0.36, [0.12–1.67])
Stand (120–180 s)			
sBP	0.52 (0.13, [0.14–1.28])	0.53 (0.13, [0.10–1.13])	0.57 (0.14, [0.27–0.96])
dBP	0.43 (0.13, [0.10–1.57])	0.44 (0.14, [0.10–1.10])	0.50 (0.19, [0.13–1.17])
MAP	0.47 (0.13, [0.06–1.50])	0.48 (0.14, [0.09–0.99])	0.51 (0.16, [0.11–1.07])
HR	0.42 (0.14, [0.01–1.06])	0.44 (0.16, [0.03–1.12])	0.47 (0.19, [0.08–0.98])
O2Hb	1.16 (0.38, [0.06–2.77])	1.16 (0.39, [0.09–2.57])	1.15 (0.39, [0.09–1.91])
HHb	0.89 (0.42, [0.07–2.80])	0.93 (0.45, [0.06–2.60])	0.97 (0.42, [0.12–2.05])
TSI	1.08 (0.36, [0.15–2.58])	1.08 (0.37, [0.15–2.78])	1.18 (0.36, [0.28–2.18])

Abbreviations: sBP: systolic blood pressure; dBP: diastolic blood pressure; MAP: mean arterial pressure; HR: heart rate; O2Hb oxygenated hemoglobin concentration; HHb deoxygenated hemoglobin concentration; TSI: tissue saturation index.

## Data Availability

The datasets generated during and/or analyzed during the current study are not publicly available due to data protection regulations but are accessible at TILDA on reasonable request. The procedures to gain access to TILDA data are specified at https://tilda.tcd.ie/data/accessing-data/.

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
