# Peer review of "Associations between Neurocardiovascular Signal Entropy and Physical Frailty"

_entropy, 2020, doi:10.3390/e23010004_

Round 1
Reviewer 1 Report
The study is interesting as it:
- discusses the question about possibility to use two relatively simple measures - approximate entropy and sample entropy as markers for physical frailty;
- Uses data from large group of people
The answer of the question from point 1 is positive and this will be of interest for many researchers and practitioners. Because of all this my opinion about the publication of the manuscript is positive.
The only note is that the average years of the people - about 64 seems to be relatively small for large percentage of physical frailness (people are still fit at such ages nowadays). It will be very interesting if this study can be done again on the same group in several years when the average years of the people will be larger. But this note is just an idea for future research.
Reviewer 2 Report
The study explores the association between complexity of the cardiovascular control and frailty using approximated entropy (ApEn) and sample entropy (SampEn). Exploration is carried out at supine rest and during active standing to probe cardiovascular control.
The study is very interesting and contains original data. However, some major issues need to be directly faced and discussed to improve visibility of the study within the current literature relevant to the assessment of complexity of cardiac and vascular controls based on spontaneous fluctuations.
- Approximate entropy (ApEn) is one of the most biased estimators of complexity based on conditional entropy calculation. If the authors planned to use it, ApEn should be corrected (see ref [45] for a suitable correction).
- Results might be discussed in relation to modifications of baroreflex control found in frailty population (see e.g. M.S.S. Buto et al, Braz J Med Biol Res, 52, e8079, 2019).
- It is well-known that complexity markers of cardiac and vascular controls carry complementary information (see A. Porta et al, J Appl Physiol, 113, 1810-1820, 2012). This issue deserves some comments in relation to the findings of the present study.
- Heart rate variability complexity is expected to decrease with pathology (see A.L. Goldberger, Lancet. 1996; 347; 1312-1314) and during orthostatic challenge (see A.M. Catai et al, Entropy, 16, 6686-6704, 2014). It is unclear whether this paradigm holds also in the present study. Conversely, trends of the complexity of vascular control with pathology might be different (see again the reference mentioned in my previous comment). Again explicit comments in relation to modifications of markers across conditions and populations might improve visibility and relevance of the present study.
- A reliable estimation of ApEn and SampEn requires stationary conditions. Conversely, nonstationarities might bias estimates of complexity based on conditional entropy estimators (see V. Magagnin et al, Physiol Meas, 32, 1775-1786, 2011). How was stationarity of the series tested? Please report the fraction of sequences fulfilling stationarity as a function of the type of considered variables.
Reviewer 3 Report
This paper tries to classify frailty group based on an entropy measure (either SampEn or ApEn) and demographic information using an unusually large dataset. I really like this paper, and just have minor suggestions for revision. The two most important are an explanation of the apparent discrepancy between Tables 2&3 and Figure 3 and an exploration of the effect of m (though of course sample size will become a problem at high m).
Introduction suggestion: For the machine learning people reading this, I'd literally say that you're trying to classify between frailty groups on the basis of some demographic variables and these entropy measures.
Line 57: "The" to "the"
Line 65-66: the way you describe ApEn is a bit confusing. Two suggestions: 1. "subset" to "trajectory" and "subsets" to "trajectories", and 2. "remain similar for subset of length m+1" to "remain similar in the next timestep"
Line 152: m=2 may be too low for this particular dataset. I'm going to ask that you explore what happens when you increase m by a bit.
Line 158: "instance time"-- just choose one.
Line 233: "fully controlled models"-- I think I know how you controlled, but I think it's worth explaining here.
Line 260-261: This is my most adamant comment. When I look at your Table 2 and Table 3, I see that ApEn has lower SD than SampEn and appears to differentiate the groups better, but just barely. As you are not controlling for age and sex in that table, I can barely believe that a fully controlled model would allow you to use ApEn or SampEn to classify. But then I look at Figure 3 and see largely excellent results. Something just doesn't track. I'm not doubting your analysis, but the discrepancy between the tables and figure should be explained.
Figure 2 caption: I cannot see a pattern in this data at all, classification-wise. But I haven't been staring at it for long. I think it would help if you described the patterns that you see in this data in the caption.
Line 351: Maybe mention other measures of entropy (like entropy rate or multi-scale entropy rate) that you have ignored, and maybe explain why you chose the entropy measures that you did. This could also go in the Introduction.
Round 2
Reviewer 2 Report
The manuscript was improved. The authors replied satisfactorily to all my issues and followed carefully the suggestions given.
Reviewer 3 Report
Thanks for changing your manuscript to incorporate the vast majority of my comments. I'm still a little worried about your statement in the Fig. 2 caption, because that's not what I see. I'm also a little worried about your statement that Tables 2-3 and Figure 3 are commensurate, because that's not what I see on a skim, but I'm perfectly willing to admit that I didn't do any serious data analysis myself and that the qualitative results one draws from your two tables might be perfectly consistent with the multivariate linear regression. I will insist on no further changes before publication.